# STICK-BREAKING VARIATIONAL AUTOENCODERS

**Eric Nalisnick**
Department of Computer Science
University of California, Irvine
`enalisni@uci.edu`

**Padhraic Smyth**
Department of Computer Science
University of California, Irvine
`smyth@ics.uci.edu`

## ABSTRACT

We extend Stochastic Gradient Variational Bayes to perform posterior inference for the weights of Stick-Breaking processes. This development allows us to define a *Stick-Breaking Variational Autoencoder* (SB-VAE), a Bayesian nonparametric version of the variational autoencoder that has a latent representation with stochastic dimensionality. We experimentally demonstrate that the SB-VAE, and a semi-supervised variant, learn highly discriminative latent representations that often outperform the Gaussian VAE's.

## 1 INTRODUCTION

Deep generative models trained via *Stochastic Gradient Variational Bayes* (SGVB) (Kingma & Welling, 2014a; Rezende et al., 2014) efficiently couple the expressiveness of deep neural networks with the robustness to uncertainty of probabilistic latent variables. This combination has lead to their success in tasks ranging from image generation (Gregor et al., 2015; Rezende et al., 2016) to semi-supervised learning (Kingma et al., 2014; Maaløe et al., 2016) to language modeling (Bowman et al., 2016). Various extensions to SGVB have been proposed (Burda et al., 2016; Maaløe et al., 2016; Salimans et al., 2015), but one conspicuous absence is an extension to Bayesian nonparametric processes. Using SGVB to perform inference for nonparametric distributions is quite attractive. For instance, SGVB allows for a broad class of non-conjugate approximate posteriors and thus has the potential to expand Bayesian nonparametric models beyond the exponential family distributions to which they are usually confined. Moreover, coupling nonparametric processes with neural network inference models equips the networks with automatic model selection properties such as a self-determined width, which we explore in this paper.

We make progress on this problem by first describing how to use SGVB for posterior inference for the weights of Stick-Breaking processes (Ishwaran & James, 2001). This is not a straightforward task as the Beta distribution, the natural choice for an approximate posterior, does not have the differentiable non-centered parametrization that SGVB requires. We bypass this obstacle by using the little-known *Kumaraswamy* distribution (Kumaraswamy, 1980).

Using the Kumaraswamy as an approximate posterior, we then reformulate two popular deep generative models—the *Variational Autoencoder* (Kingma & Welling, 2014a) and its semi-supervised variant (model M2 proposed by Kingma et al. (2014))—into their nonparametric analogs. These models perform automatic model selection via an infinite capacity hidden layer that employs as many stick segments (latent variables) as the data requires. We experimentally show that, for datasets of natural images, stick-breaking priors improve upon previously proposed deep generative models by having a latent representation that better preserves class boundaries and provides beneficial regularization for semi-supervised learning.

## 2 BACKGROUND

We begin by reviewing the relevant background material on Variational Autoencoders (Kingma & Welling, 2014a), Stochastic Gradient Variational Bayes (also known as *Stochastic Backpropagation*) (Kingma & Welling, 2014a; Rezende et al., 2014), and Stick-Breaking Processes (Ishwaran & James, 2001).

## 2.1 Variational Autoencoders

A *Variational Autoencoder* (VAE) is model comprised of two multilayer perceptrons: one acts as a *density network* (MacKay & Gibbs, 1999) mapping a latent variable $\mathbf{z}_i$ to an observed datapoint $\mathbf{x}_i$, and the other acts as an *inference model* (Salimans & Knowles, 2013) performing the reverse mapping from $\mathbf{x}_i$ to $\mathbf{z}_i$. Together the two form a computational pipeline that resembles an unsupervised autoencoder (Hinton & Salakhutdinov, 2006). The generative process can be written mathematically as

$$\mathbf{z}_i \sim p(\mathbf{z}) \, , \mathbf{x}_i \sim p_{\boldsymbol{\theta}}(\mathbf{x}|\mathbf{z}_i) \tag{1}$$

where $p(\mathbf{z})$ is the prior and $p_{\boldsymbol{\theta}}(\mathbf{x}|\mathbf{z}_i)$ is the density network with parameters $\boldsymbol{\theta}$. The approximate posterior of this generative process, call it $q_{\boldsymbol{\phi}}(\mathbf{z}|\mathbf{x}_i)$, is then parametrized by the inference network (with parameters $\boldsymbol{\phi}$). In previous work (Kingma & Welling, 2014a; Rezende & Mohamed, 2015; Burda et al., 2016; Li & Turner, 2016), the prior $p(\mathbf{z})$ and variational posterior have been marginally Gaussian.

## 2.2 Stochastic Gradient Variational Bayes

The VAE's generative and variational parameters are estimated by *Stochastic Gradient Variational Bayes* (SGVB). SGVB is distinguished from classical variational Bayes by it's use of differentiable Monte Carlo (MC) expectations. To elaborate, consider SGVB's approximation of the usual evidence lowerbound (ELBO) (Jordan et al., 1999):

$$\tilde{\mathcal{L}}(\boldsymbol{\theta}, \boldsymbol{\phi}; \mathbf{x}_i) = \frac{1}{S} \sum_{s=1}^{S} \log p_{\boldsymbol{\theta}}(\mathbf{x}_i|\hat{\mathbf{z}}_{i,s}) - KL(q_{\boldsymbol{\phi}}(\mathbf{z}_i|\mathbf{x}_i)||p(\mathbf{z})) \tag{2}$$

for $S$ samples of $\mathbf{z}_i$ and where $KL$ is the Kullback-Leibler divergence. An essential requirement of SGVB is that the latent variable be represented in a *differentiable, non-centered parametrization* (DNCP) (Kingma & Welling, 2014b); this is what allows the gradients to be taken through the MC expectation, i.e.:

$$\frac{\partial}{\partial \boldsymbol{\phi}} \sum_{s=1}^{S} \log p_{\boldsymbol{\theta}}(\mathbf{x}_i|\hat{\mathbf{z}}_{i,s}) = \sum_{s=1}^{S} \frac{\partial}{\partial \hat{\mathbf{z}}_{i,s}} \log p_{\boldsymbol{\theta}}(\mathbf{x}_i|\hat{\mathbf{z}}_{i,s}) \frac{\partial \hat{\mathbf{z}}_{i,s}}{\partial \boldsymbol{\phi}}.$$

In other words, $\mathbf{z}$ must have a functional form that deterministically exposes the variational distribution's parameters and allows the randomness to come from draws from some fixed distribution. Location-scale representations and inverse cumulative distribution functions are two examples of DNCPs. For instance, the VAE's Gaussian latent variable (with diagonal covariance matrix) is represented as $\hat{\mathbf{z}}_i = \boldsymbol{\mu} + \boldsymbol{\sigma} \odot \boldsymbol{\epsilon}$ where $\boldsymbol{\epsilon} \sim \mathrm{N}(\mathbf{0}, \mathbb{1})$.

## 2.3 Stick-Breaking Processes

Lastly, we define *stick-breaking processes* with the ultimate goal of using their weights for the VAE's prior $p(\mathbf{z})$. A random measure is referred to as a *stick-breaking prior* (SBP) (Ishwaran & James, 2001) if it is of the form $G(\cdot) = \sum_{k=1}^{\infty} \pi_k \delta_{\zeta_k}$ where $\delta_{\zeta_k}$ is a discrete measure concentrated at $\zeta_k \sim G_0$, a draw from the base distribution $G_0$ (Ishwaran & James, 2001). The $\pi_k$s are random weights independent of $G_0$, chosen such that $0 \le \pi_k \le 1$, and $\sum_k \pi_k = 1$ almost surely. SBPs have been termed as such because of their constructive definition known as the *stick-breaking process* (Sethuraman, 1994). Mathematically, this definition implies that the weights can be drawn according to the following iterative procedure:

$$\pi_k = \begin{cases} v_1 \text{ if } k = 1 \\ v_k \prod_{j<k}(1 - v_j) \text{ for } k > 1 \end{cases} \tag{3}$$

where $v_k \sim \mathrm{Beta}(\alpha, \beta)$. When $v_k \sim \mathrm{Beta}(1, \alpha_0)$, then we have the stick-breaking construction for the *Dirichlet Process* (Ferguson, 1973). In this case, the name for the joint distribution over the infinite sequence of stick-breaking weights is the Griffiths, Engen and McCloskey distribution with concentration parameter $\alpha_0$ (Pitman, 2002): $(\pi_1, \pi_2, \dots) \sim \mathrm{GEM}(\alpha_0)$.

## 3 SGVB FOR GEM RANDOM VARIABLES

Having covered the relevant background material, we now discuss the first contribution of this paper, using Stochastic Gradient Variational Bayes for the weights of a stick-breaking process. Inference for the random measure $G(\cdot)$ is an open problem that we leave to future work. We focus on performing inference for just the series of stick-breaking weights, which we will refer to as GEM random variables after their joint distribution.

### 3.1 COMPOSITION OF GAMMA RANDOM VARIABLES

In the original SGVB paper, Kingma & Welling (2014a) suggest representing the Beta distribution as a composition of Gamma random variables by using the fact $v \sim \text{Beta}(\alpha, \beta)$ can be sampled by drawing Gamma variables $x \sim \text{Gamma}(\alpha, 1)$, $y \sim \text{Gamma}(\beta, 1)$ and composing them as $v = x/(x + y)$. However, this representation still does not admit a DNCP as the Gamma distribution does not have one with respect to its shape parameter. Knowles (2015) suggests that when the shape parameter is near zero, the following asymptotic approximation of the inverse CDF is a suitable DNCP:

$$F^{-1}(\hat{u}) \approx \frac{(\hat{u}a\Gamma(a))^{\frac{1}{a}}}{b} \tag{4}$$

for $\hat{u} \sim \text{Uniform}(0, 1)$, shape parameter $a$, and scale parameter $b$. This approximation becomes poor as $a$ increases, however, and Knowles recommends a finite difference approximation of the inverse CDF when $a \geq 1$.

### 3.2 THE KUMARASWAMY DISTRIBUTION

Another candidate posterior is the little-known *Kumaraswamy* distribution (Kumaraswamy, 1980). It is a two-parameter continuous distribution also on the unit interval with a density function defined as

$$\text{Kumaraswamy}(x; a, b) = abx^{a-1}(1 - x^a)^{b-1} \tag{5}$$

for $x \in (0, 1)$ and $a, b > 0$. In fact, if $a = 1$ or $b = 1$ or both, the Kumaraswamy and Beta are equivalent, and for equivalent parameter settings, the Kumaraswamy resembles the Beta albeit with higher entropy. The DNCP we desire is the Kumaraswamy's closed-form inverse CDF. Samples can be drawn via the inverse transform:

$$x \sim (1 - u^{\frac{1}{b}})^{\frac{1}{a}} \text{ where } u \sim \text{Uniform}(0, 1). \tag{6}$$

Not only does the Kumaraswamy make sampling easy, its KL-divergence from the Beta can be closely approximated in closed-form (for ELBO computation).

#### 3.2.1 GAUSS-LOGIT PARAMETRIZATION

Another promising parametrization is inspired by the *Probit Stick-Breaking Process* (Rodriguez & Dunson, 2011). In a two-step process, we can draw a Gaussian and then use a squashing function to map it on $(0, 1)$:

$$\hat{v}_k = g(\mu_k + \sigma_k \epsilon) \tag{7}$$

where $\epsilon \sim \text{N}(0, 1)$. In the Probit SBP, $g(\cdot)$ is taken to be the Gaussian CDF, and it is chosen as such for posterior sampling considerations. This choice is impractical for our purposes, however, since the Gaussian CDF does not have a closed form. Instead, we use the logistic function $g(x) = 1/(1 + e^{-x})$.

## 4 STICK-BREAKING VARIATIONAL AUTOENCODERS

Given the discussion above, we now propose the following novel modification to the VAE. Instead of drawing the latent variables from a Gaussian distribution, we draw them from the GEM distribution, making the hidden representation an infinite sequence of stick-breaking weights. We term this model a *Stick-Breaking Variational Autoencoder* (SB-VAE) and below detail the generative and inference processes implemented in the decoding and encoding models respectively.

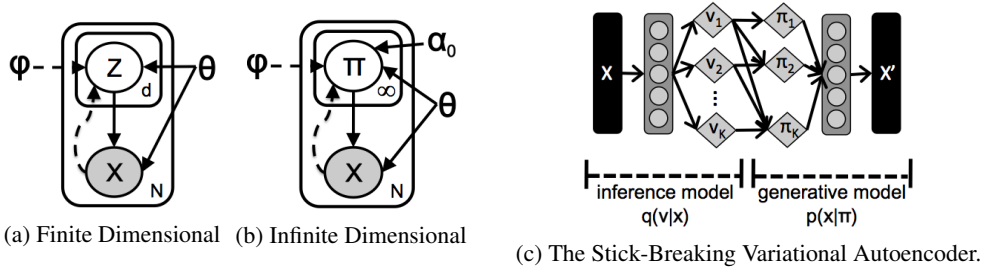

(a) Finite Dimensional   (b) Infinite Dimensional

(c) The Stick-Breaking Variational Autoencoder.

Figure 1: Subfigures (a) and (b) show the plate diagrams for the relevant latent variable models. Solid lines denote the generative process and dashed lines the inference model. Subfigure (a) shows the finite dimensional case considered in (Kingma & Welling, 2014a), and (b) shows the infinite dimensional case of our concern. Subfigure (c) shows the feedforward architecture of the *Stick-Breaking Autoencoder*, which is a neural-network-based parametrization of the graphical model in (b).

## 4.1   GENERATIVE PROCESS

The generative process is nearly identical to previous VAE formulations. The crucial difference is that we draw the latent variable from a stochastic process, the GEM distribution. Mathematically, the hierarchical formulation is written as

$$\boldsymbol{\pi}_i \sim \text{GEM}(\alpha_0) \, , \, \mathbf{x}_i \sim p_{\boldsymbol{\theta}}(\mathbf{x}_i | \boldsymbol{\pi}_i) \tag{8}$$

where $\boldsymbol{\pi}_i$ is the vector of stick-breaking weights and $\alpha_0$ is the concentration parameter of the GEM distribution. The likelihood model $p_{\boldsymbol{\theta}}(\mathbf{x}_i | \boldsymbol{\pi}_i)$ is a density network just as described in Section 2.1.

## 4.2   INFERENCE

The inference process—how to draw $\boldsymbol{\pi}_i \sim q_{\phi}(\boldsymbol{\pi}_i | \mathbf{z}_i)$—requires modification from the standard VAE's in order to sample from the GEM's stick-breaking construction. Firstly, an inference network computes the parameters of $K$ fraction distributions and samples values $v_{i,k}$ according to one of the parametrizations in Section 3. Next, a linear-time operation composes the stick segments from the sampled fractions:

$$\boldsymbol{\pi}_i = (\pi_{i,1}, \pi_{i,2}, \dots, \pi_{i,K}) = \left( v_{i,1}, v_{i,2}(1 - v_{i,1}), \dots, \prod_{j=1}^{K-1}(1 - v_{i,j}) \right). \tag{9}$$

The computation path is summarized in Figure 1 (c) with arrows denoting the direction of feedforward computation. The gray blocks represent any deterministic function that can be trained with gradient descent—i.e. one or more neural network layers. Optimization of the SB-VAE is done just as for the VAE, by optimizing Equation 2 w.r.t. $\phi$ and $\boldsymbol{\theta}$. The KL divergence term can be computed (or closely approximated) in closed-form for all three parametrizations under consideration; the Kumaraswamy-to-Beta KL divergence is given in the appendix.

An important detail is that the $K$th fraction $v_{i,K}$ is always set to one to ensure the stick segments sum to one. This truncation of the variational posterior does *not* imply that we are using a finite dimensional prior. As explained by Blei & Jordan (2006), the truncation level is a variational parameter and not part of the prior model specification. Truncation-free posteriors have been proposed, but these methods use split-and-merge steps (Hughes et al., 2015) or collapsed Gibbs sampling, both of which are not applicable to the models we consider. Nonetheless, because SGVB imposes few limitations on the inference model, it is possible to have an untruncated posterior. We conducted exploratory experiments using a truncation-free posterior by adding extra variational parameters in an on-line fashion, initializing new weights if more than 1% of the stick remained unbroken. However, we found this made optimization slower without any increase in performance.

## 5 SEMI-SUPERVISED MODEL

We also propose an analogous approach for the semi-supervised relative of the VAE, the M2 model described by Kingma et al. (2014). A second latent variable $y_i$ is introduced that represents a class label. Its distribution is the categorical one: $q_\phi(y_i|\mathbf{x}_i) = \text{Cat}(y|g_y(\mathbf{x}_i))$ where $g_y$ is a non-linear function of the inference network. Although $y$'s distribution is written as independent of $\mathbf{z}$, the two share parameters within the inference network and thus act to regularize one another. We assume the same factorization of the posterior and use the same objectives as in the finite dimensional version (Kingma et al., 2014). Since $y_i$ is present for some but not all observations, semi-supervised DGMs need to be trained with different objectives depending on whether the label is present or not. If the label is present, following Kingma et al. (2014) we optimize

$$\tilde{\mathcal{J}}(\boldsymbol{\theta},\boldsymbol{\phi};\mathbf{x}_i,y_i) = \frac{1}{S}\sum_{s=1}^{S}\log p_{\boldsymbol{\theta}}(\mathbf{x}_i|\boldsymbol{\pi}_{i,s},y_i) - KL(q_\phi(\boldsymbol{\pi}_i|\mathbf{x}_i)||p(\boldsymbol{\pi}_i;\boldsymbol{\alpha}_0)) + \log q_\phi(y_i|\mathbf{x}_i) \quad (10)$$

where $\log q_\phi(y_i|\mathbf{x}_i)$ is the log-likelihood of the label. And if the label is missing, we optimize

$$\tilde{\mathcal{J}}(\boldsymbol{\theta},\boldsymbol{\phi};\mathbf{x}_i) = \frac{1}{S}\sum_{s=1}^{S}\sum_{y_j} q_\phi(y_j|\mathbf{x}_i)\left[\log p_{\boldsymbol{\theta}}(\mathbf{x}_i|\boldsymbol{\pi}_{i,s},y_j)\right] + \mathbb{H}[q_\phi(y|\mathbf{x}_i)]$$
$$- KL(q_\phi(\boldsymbol{\pi}_i|\mathbf{x}_i)||p(\boldsymbol{\pi}_i;\boldsymbol{\alpha}_0)) \quad (11)$$

where $\mathbb{H}[q_\phi(y_i|\mathbf{x}_i)]$ is the entropy of $y$'s variational distribution.

## 6 RELATED WORK

To the best of our knowledge, neither SGVB nor any of the other recently proposed amortized VI methods (Kingma & Welling, 2014b; Rezende & Mohamed, 2015; Rezende et al., 2014; Tran et al., 2016) have been used in conjunction with BNP priors. There has been work on using nonparametric posterior approximations—in particular, the *Variational Gaussian Process* (Tran et al., 2016)—but in that work the variational distribution is nonparametric, not the generative model. Moreover, we are not aware of prior work that uses SGVB for Beta (or Beta-like) random variables[1].

In regards to the autoencoder implementations we describe, they are closely related to the existing work on representation learning with adaptive latent factors—i.e. where the number of latent dimensions grows as the data necessitates. The best known model of this kind is the infinite binary latent feature model defined by the Indian Buffet Process (Ghahramani & Griffiths, 2005); but its discrete latent variables prevent this model from admitting fully differentiable inference. Recent work that is much closer in spirit is the *Infinite Restricted Boltzmann Machine* (iRBM) (Côté & Larochelle, 2016), which has gradient-based learning, expands its capacity by adding hidden units, and induces a similar ordering on latent factors. The most significant difference between our SB-VAE and the iRBM is that the latter's nonparametric behavior arises from a particular definition of the energy function of the Gibbs distribution, not from an infinite dimensional Bayesian prior. Lastly, our training procedure bears some semblance to *Nested Dropout* (Rippel et al., 2014), which removes all hidden units at an index lower than some threshold index. The SB-VAE can be seen as performing soft nested dropout since the latent variable values decrease as their index increases.

## 7 EXPERIMENTS

We analyze the behavior of the three parametrizations of the SB-VAE and examine how they compare to the Gaussian VAE. We do this by examining their ability to reconstruct the data (i.e. density estimation) and to preserve class structure. Following the original DGM papers (Kingma et al., 2014; Kingma & Welling, 2014a; Rezende et al., 2014), we performed unsupervised and semi-supervised

---

[1]During preparation of this draft, the work of Ruiz et al. (2016) on the *Generalized Reparametrization Gradient* was released (on 10/7/16), which can be used for Beta random variables. We plan to compare their technique to our proposed use of the Kumaraswamy in subsequent drafts.

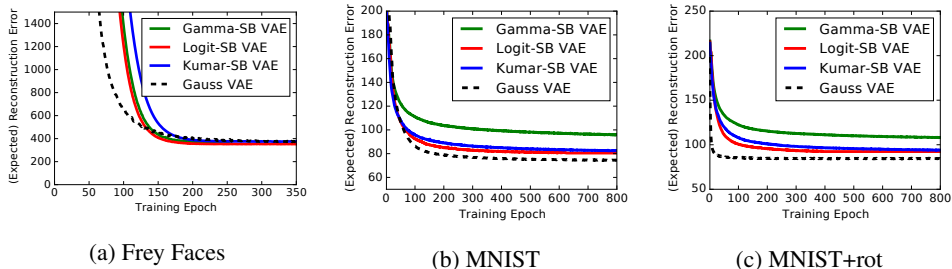

(a) Frey Faces

(b) MNIST

(c) MNIST+rot

Figure 2: Subfigure (a) shows test (expected) reconstruction error vs training epoch for the SB-VAE and Gauss VAE on the Frey Faces dataset, subfigure (b) shows the same quantities for the same models on the MNIST dataset, and subfigure (c) shows the same quantities for the same models on the MNIST+rot dataset.

tasks on the following image datasets: Frey Faces[2], MNIST, MNIST+rot, and Street View House Numbers[3] (SVHN). MNIST+rot is a dataset we created by combining MNIST and rotated MNIST[4] for the purpose of testing the latent representation under the conjecture that the rotated digits should use more latent variables than the non-rotated ones.

Complete implementation and optimization details can be found in the appendix and code repository[5]. In all experiments, to best isolate the effects of Gaussian versus stick-breaking latent variables, the same architecture and optimization hyperparameters were used for each model. The only difference was in the prior: $p(\mathbf{z}) = \mathrm{N}(\mathbf{0}, \mathbb{1})$ for Gaussian latent variables and $p(v) = \mathrm{Beta}(1, \alpha_0)$ (Dirichlet process) for stick-breaking latent variables. We cross-validated the concentration parameter over the range $\alpha_0 \in \{1, 3, 5, 8\}$. The Gaussian model's performance potentially could have been improved by cross validating its prior variance. However, the standard Normal prior is widely used as a default choice (Bowman et al., 2016; Gregor et al., 2015; Kingma et al., 2014; Kingma & Welling, 2014a; Rezende et al., 2014; Salimans et al., 2015), and our goal is to experimentally demonstrate a stick-breaking prior is a competitive alternative.

## 7.1 UNSUPERVISED

We first performed unsupervised experiments testing each model's ability to recreate the data as well as preserve class structure (without having access to labels). The inference and generative models both contained one hidden layer of 200 units for Frey Faces and 500 units for MNIST and MNIST+rot. For Frey Faces, the Gauss VAE had a 25 dimensional (factorized) distribution, and we set the truncation level of the SB-VAE also to $K = 25$, so the SB-VAE could use only as many latent variables as the Gauss VAE. For the MNIST datasets, the latent dimensionality/truncation-level was set at 50. Cross-validation chose $\alpha_0 = 1$ for Frey Faces and $\alpha_0 = 5$ for both MNISTs.

**Density Estimation.** In order to show each model's optimization progress, Figure 2 (a), (b), and (c) report test expected reconstruction error (i.e. the first term in the ELBO) vs training progress (epochs) for Frey Faces, MNIST, and MNIST+rot respectively. Optimization proceeds much the same in both models except that the SB-VAE learns at a slightly slower pace for all parametrizations. This is not too surprising since the recursive definition of the latent variables likely causes coupled gradients.

We compare the final converged models in Table 1, reporting the marginal likelihood of each model via the MC approximation $\log p(\mathbf{x}_i) \approx \log \frac{1}{S} \sum_s p(\mathbf{x}_i | \hat{\mathbf{z}}_{i,s}) p(\hat{\mathbf{z}}_{i,s}) / q(\hat{\mathbf{z}}_{i,s})$ using 100 samples. The Gaussian VAE has a better likelihood than all stick-breaking implementations ($\sim 96$ vs $\sim 98$). Between the stick-breaking parametrizations, the Kumaraswamy outperforms both the Gamma and Gauss-Logit on both datasets, which is not surprising given the others' flaws (i.e. the Gamma is approximate, the Gauss-Logit is restricted). Given this result, we used the Kumaraswamy parametriza-

---

[2]Available at http://www.cs.nyu.edu/~roweis/data.html

[3]Available at http://ufldl.stanford.edu/housenumbers/

[4]Available at http://www.iro.umontreal.ca/~lisa/twiki/bin/view.cgi/Public/MnistVariations

[5]Theano implementations available at https://github.com/enalisnick/stick-breaking_dgms

| Model | $-\log p(\mathbf{x}_i)$ | |
| --- | --- | --- |
| | **MNIST** | **MNIST+rot** |
| Gauss VAE | 96.80 | 108.40 |
| Kumar-SB VAE | 98.01 | 112.33 |
| Logit-SB VAE | 99.48 | 114.09 |
| Gamma-SB VAE | 100.74 | 113.22 |

Table 1: Marginal likelihood results (estimated) for Gaussian VAE and the three parametrizations of the Stick-Breaking VAE.

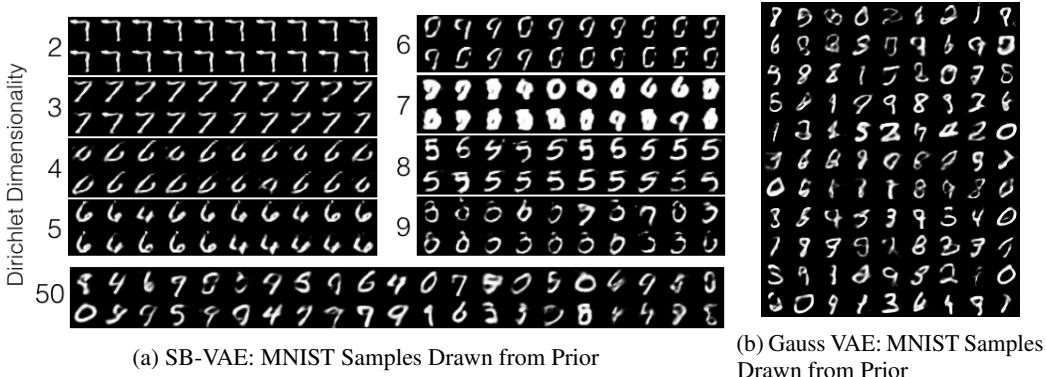

(a) SB-VAE: MNIST Samples Drawn from Prior

(b) Gauss VAE: MNIST Samples Drawn from Prior

Figure 3: Subfigure (a) depicts samples from the SB-VAE trained on MNIST. We show the ordered, factored nature of the latent variables by sampling from Dirichlet's of increasing dimensionality. Subfigure (b) depicts samples from the Gauss VAE trained on MNIST.

tion for all subsequently reported experiments. Note that the likelihoods reported are worse than the ones reported by Burda et al. (2016) because our training set consisted of 50k examples whereas theirs contained 60k (training and validation).

We also investigated whether the SB-VAE is using its adaptive capacity in the manner we expect, i.e., the SB-VAE should use a larger latent dimensionality for the rotated images in MNIST+rot than it does for the non-rotated ones. We examined if this is the case by tracking how many 'breaks' it took the model to deconstruct 99% of the stick. On average, the rotated images in the training set were represented by $28.7$ dimensions and the non-rotated by $27.4$. Furthermore, the rotated images used more latent variables in eight out of ten classes. Although the difference is not as large as we were expecting, it is statistically significant. Moreover, the difference is made smaller by the non-rotated one digits, which use 32 dimensions on average, the most for any class. The non-rotated average decreases to 26.3 when ones are excluded.

Figure 3 (a) shows MNIST digits drawn from the SB-VAE by sampling from the prior—i.e. $v_k \sim$ Beta$(1, 5)$, and Figure 3 (b) shows Gauss VAE samples for comparison. SB-VAE samples using all fifty dimensions of the truncated posterior are shown in the bottom block. Samples from Dirichlets constrained to a subset of the dimensions are shown in the two columns in order to test that the latent features are concentrating onto lower-dimensional simplices. This is indeed the case: adding a latent variable results in markedly different but still coherent samples. For instance, the second and third dimensions seem to capture the 7-class, the fourth and fifth the 6-class, and the eighth the 5-class. The seventh dimension seems to model notably thick digits.

**Discriminative Qualities.** The discriminative qualities of the models' latent spaces are assessed by running a k-Nearest Neighbors classifier on (sampled) MNIST latent variables. Results are shown in the table in Figure 4 (a). The SB-VAE exhibits conspicuously better performance than the Gauss VAE at all choices of $k$, which suggests that although the Gauss VAE converges to a better likelihood, the SB-VAE's latent space better captures class structure. We also report results for two Gaussian *mixture* VAEs: Dilokthanakul et al. (2016)'s *Gaussian mixture Variational Autoencoder* (GMVAE)

| | k=3 | k=5 | k=10 |
|---|---|---|---|
| SB-VAE | 9.34 | 8.65 | 8.90 |
| DLGMM | 9.14 | 8.38 | 8.42 |
| Gauss VAE | 28.4 | 20.96 | 15.33 |
| Raw Pixels | 2.95 | 3.12 | 3.35 |
| GMVAE[6] | — | 8.96 | — |

(a) MNIST: Test error for kNN on latent space

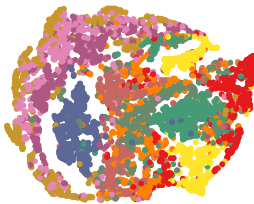

(b) MNIST SB-VAE

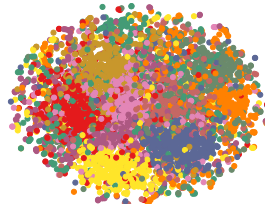

(c) MNIST Gauss VAE

Figure 4: Subfigure (a) shows results of a kNN classifier trained on the latent representations produced by each model. Subfigures (b) and (c) show t-SNE projections of the latent representations learned by the SB-VAE and Gauss VAE respectively.

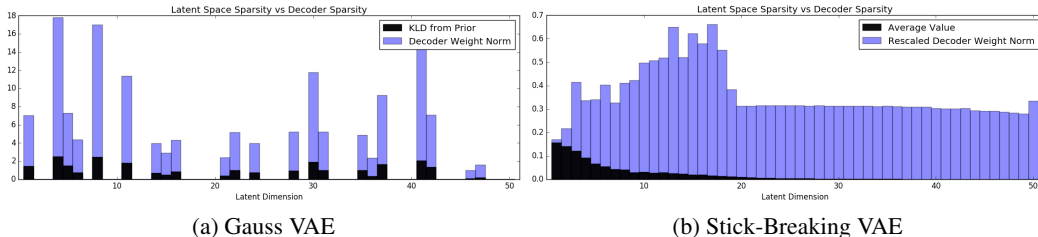

(a) Gauss VAE

(b) Stick-Breaking VAE

Figure 5: Sparsity in the latent representation vs sparsity in the decoder network. The Gaussian VAE 'turns off' unused latent dimensions by setting the outgoing weights to zero (in order to dispel the sampled noise). The SB VAE, on the other hand, also has sparse representations but without decay of the associated decoder weights.

and Nalisnick et al. (2016)'s *Deep Latent Gaussian Mixture Model* (DLGMM). The GMVAE[6] has sixteen mixture components and the DLGMM has five, and hence both have many more parameters than the SB-VAE. Despite the SB-VAE's lower capacity, we see that its performance is competitive to the mixture VAEs' (8.65 vs 8.38/8.96).

The discriminative qualities of the SB-VAE's latent space are further supported by Figures 4 (b) and (c). t-SNE was used to embed the Gaussian (c) and stick-breaking (b) latent MNIST representations into two dimensions. Digit classes (denoted by color) in the stick-breaking latent space are clustered with noticeably more cohesion and separation.

**Combating Decoder Pruning.** The 'component collapsing' behavior of the variational autoencoder has been well noted (Maaløe et al., 2016): the model will set to zero the outgoing weights of latent variables that remain near the prior. Figure 5 (a) depicts this phenomenon for the Gauss VAE by plotting the KL divergence from the prior and outgoing decoder weight norm for each latent dimension. We see the weights are only nonzero in the dimensions in which there is posterior deviation. Ostensibly the model receives only sampling noise from the dimensions that remain at the prior, and setting the decoder weights to zero quells this variance. While the behavior of the Gauss VAE is not necessarily improper, all examples are restricted to pass through the same latent variables. A sparse-coded representation—one having few active components per example (like the Gauss VAE) but diversity of activations across examples–would likely be better.

We compare the activation patterns against the sparsity of the decoder for the SB-VAE in Figure 5 (b). Since KL-divergence doesn't directly correspond to sparsity in stick-breaking latent variables like it does for Gaussian ones, the black lines denote the average activation value per dimension. Similarly to (a), blue lines denoted the decoder weight norms, but they had to be down-scaled by a factor of 100 so they could be visualized on the same plot. The SB-VAE does not seem to have any component collapsing, which is not too surprising since the model can set latent variables to zero to deactivate decoder weights without being in the heart of the prior. We conjecture that this increased capacity is

---

[6]The GMVAE's evaluation is different from performing kNN. Rather, test images are assigned to clusters and whole clusters are given a label. Thus results are not strictly comparable but the ultimate goal of unsupervised MNIST classification is the same.

| | MNIST (N=45,000) | | | MNIST+rot (N=70,000) | | | SVHN (N=65,000) | | |
|---|---|---|---|---|---|---|---|---|---|
| | 10% | 5% | 1% | 10% | 5% | 1% | 10% | 5% | 1% |
| SB-DGM | $4.86_{\pm.14}$ | $5.29_{\pm.39}$ | $\mathbf{7.34}_{\pm.47}$ | $\mathbf{11.78}_{\pm.39}$ | $\mathbf{14.27}_{\pm.58}$ | $\mathbf{27.67}_{\pm1.39}$ | $\mathbf{32.08}_{\pm4.00}$ | $\mathbf{37.07}_{\pm5.22}$ | $\mathbf{61.37}_{\pm3.60}$ |
| Gauss-DGM | $\mathbf{3.95}_{\pm.15}$ | $\mathbf{4.74}_{\pm.43}$ | $11.55_{\pm2.28}$ | $21.78_{\pm.73}$ | $27.72_{\pm.69}$ | $38.13_{\pm.95}$ | $36.08_{\pm1.49}$ | $48.75_{\pm1.47}$ | $69.58_{\pm1.64}$ |
| kNN | $6.13_{\pm.13}$ | $7.66_{\pm.10}$ | $15.27_{\pm.76}$ | $18.41_{\pm.01}$ | $23.43_{\pm.01}$ | $37.98_{\pm.01}$ | $64.81_{\pm.34}$ | $68.94_{\pm.47}$ | $76.64_{\pm.54}$ |

Table 2: Percent error on three semi-supervised classification tasks with 10%, 5%, and 1% of labels present for training. Our DGM with stick-breaking latent variables (SB-DGM) is compared with a DGM with Gaussian latent variables (Gauss-DGM), and a k-Nearest Neighbors classifier (k=5).

the reason stick-breaking variables demonstrate better discriminative performance in many of our experiments.

## 7.2 SEMI-SUPERVISED

We also performed semi-supervised classification, replicating and extending the experiments in the original semi-supervised DGMs paper (Kingma et al., 2014). We used the MNIST, MNIST+rot, and SVHN datasets and reduced the number of labeled training examples to 10%, 5%, and 1% of the total training set size. Labels were removed completely at random and as a result, class imbalance was all but certainly introduced. Similarly to the unsupervised setting, we compared DGMs with stick-breaking (SB-DGM) and Gaussian (Gauss-DGM) latent variables against one another and a baseline k-Nearest Neighbors classifier (k=5). We used 50 for the latent variable dimensionality / truncation level. The MNIST networks use one hidden layer of 500 hidden units. The MNIST+rot and SVHN networks use four hidden layers of 500 units in each. The last three hidden layers have identity function skip-connections. Cross-validation chose $\alpha_0 = 5$ for MNISTs and $\alpha_0 = 8$ for SVHN.

**Quantitative Evaluation.** Table 2 shows percent error on a test set when training with the specified percentage of labeled examples. We see the the SB-DGM performs markedly better across almost all experiments. The Gauss-DGM achieves a superior error rate only on the easiest tasks: MNIST with 10% and 5% of the data labeled.

## 8 CONCLUSIONS

We have described how to employ the Kumaraswamy distribution to extend Stochastic Gradient Variational Bayes to the weights of stick-breaking Bayesian nonparametric priors. Using this development we then defined deep generative models with infinite dimensional latent variables and showed that their latent representations are more discriminative than those of the popular Gaussian variant. Moreover, the only extra computational cost is in assembling the stick segments, a linear operation on the order of the truncation size. Not only are the ideas herein immediately useful as presented, they are an important first-step to integrating black box variational inference and Bayesian nonparametrics, resulting in scalable models that have differentiable control of their capacity. In particular, applying SGVB to full Dirichlet processes with non-trivial base measures is an interesting next step. Furthermore, differentiable stick-breaking has the potential to increase the dynamism and adaptivity of neural networks, a subject of recent interest (Graves, 2016), in a probabilistically principled way.

## ACKNOWLEDGEMENTS

Many thanks to Marc-Alexandre Côté and Hugo Larochelle for helpful discussions. This work was supported in part by NSF award number IIS-1320527.

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

## APPENDIX

### KUMARASWAMY-BETA KL-DIVERGENCE

The Kullback-Leibler divergence between the Kumaraswamy and Beta distributions is

$$
\mathbb{E}_q \left[ \log \frac{q(v)}{p(v)} \right] = \frac{a - \alpha}{a} \left( -\gamma - \Psi(b) - \frac{1}{b} \right) + \log ab + \log B(\alpha, \beta) -
$$
$$
\frac{b-1}{b} + (\beta - 1)b \sum_{m=1}^{\infty} \frac{1}{m + ab} B \left( \frac{m}{a}, b \right) \tag{12}
$$

where $q$ is Kumaraswamy($a$,$b$), $p$ is Beta($\alpha$,$\beta$). Above $\gamma$ is Euler's constant, $\Psi(\cdot)$ is the Digamma function, and $B(\cdot)$ is the Beta function. The infinite sum is present in the KLD because a Taylor expansion is needed to represent $\mathbb{E}_q[\log(1 - v_k)]$; it should be well approximated by the first few terms.

### EXPERIMENTS AND OPTIMIZATION

Below we give more details about our experiments and model optimization. Also we have released Theano implementations available at `https://github.com/enalisnick/stick-breaking_dgms`. All experiments were run on AWS G2.2XL instances.

In regards to datasets, we used the following train-valid-test splits. Frey Faces was divided into $\{1500, 100, 300\}$, MNIST into $\{45000, 5000, 10000\}$, MNIST+rot into $\{70000, 10000, 20000\}$, and SVHN into $\{65000, 8257, 26032\}$. SVHN was the only dataset that underwent preprocessing; following (Kingma et al., 2014), we reduced the dimensionality via PCA to 500 dimensions that capture 99.9% of the data's variance. No effort was made to preserve class balance across train-valid splits nor during label removal for the semi-supervised tasks.

Regarding optimization, all models were trained with minibatches of size 100 and using *AdaM* (Kingma & Ba, 2014) to set the gradient descent step size. For AdaM, we used $\alpha = 0.0003$, $b1 = 0.95$, and $b2 = 0.999$ in all experiments. Early stopping was used during semi-supervised training with a look-ahead threshold of 30 epochs. For the semi-supervised deep generative models, classification loss needs up-weighted in some way. In (Kingma et al., 2014), an extra weight was placed on the label log likelihood term. We attempted this strategy but attained better performance (for all models) by re-weighting the contribution of the supervised data within each mini-batch, i.e. $\lambda \nabla \tilde{\mathcal{J}}(\boldsymbol{\theta}, \boldsymbol{\phi}; \mathbf{x}_i, y_i) + (1 - \lambda) \tilde{\mathcal{J}}(\boldsymbol{\theta}, \boldsymbol{\phi}; \mathbf{x}_i)$. We calibrated $\lambda$ by comparing the log likelihood of the supervised and unsupervised data in preliminary training runs and setting the parameter such that the supervised data had a slightly higher likelihood. For the MNIST datasets, $\lambda = .375$, and for SVHN, $\lambda = .45$.

As for the model architectures, all experiments used ReLUs exclusively for hidden unit activations. The dimensionality / truncation-level of the latent variables was set at 50 for every experiment except Frey Faces. All weights were initialized by drawing from N($\mathbf{0}, 0.001 \cdot \mathbb{1}$), and biases were set to zero to start. No regularization (dropout, weight decay, etc) was used, and only one sample was used for each calculation of the Monte Carlo expectations. We used the leading ten terms to compute the infinite sum in the KL divergence between the Beta and Kumaraswamy.

