# Peer review of "Stick-Breaking Variational Autoencoders"

_ICLR 2017 — accepted_

[Public Comment · Christian A Naesseth · 07 Nov 2016]
**Reparameterization for Beta**

Very interesting work! 

Shameless plug: I would suggest to look at our recent paper on reparameterization for random variables drawn using rejection sampling

[Official Review · AnonReviewer2 · rating 4 · confidence 4 · 16 Dec 2016 (modified: 17 Dec 2016)]
**No Title**

The paper attempts to combine Variational Auto-Encoders with the Stick-Breaking process. The motivation is to tackle the component collapsing and have a representation with stochastic dimensionality. To demonstrate the merit of their approach, the authors test this model on MNIST and SVHN in an unsupervised and semi-supervised fashion.
After reading the paper in more detail, I find that the claim that the dimensionality of the latent variable is stochastic does not seem quite correct: all latent variables are "used" (which actually enable backpropagation) but the latent variables are parametrized differently (into $\pi$) and the decoding process is altered as to give the impression of sparsity. The way all these latent variables are used does not involve any marginalization but is very similar to the common soft-gating mechanism already used in LSTM or attentional model.
With respect to the Figure 5b showing the decoder input weights: component collapsing probably does not have the same effect as Gaussian prior. $\pi$ is positive therefore having a very small average value might mean that its value is close to zero most of the time, not requiring any update on the weight. For the standard Gaussian prior, component collapsing means having a very noisy input with no signal involved, which forces the decoder to shut down this channel, i.e. have small incoming weights from this collapsed variable.
Adding a histogram of the latent variables in addition to that might help decide if the associated weights are relatively large because they are actually used or if it's because the inputs are zero anyway.
The semi-supervised results are better than a weaker version of the model used in (Kingma et al., 2014), but as to have a fairer comparison, the results should be compared with the M1+M2 model in that paper, even if that requires also using two VAEs.

[Public Comment · Zhenwen Dai · 16 Dec 2016]
**Motivation for Stick-Breaking Process**

Very Interesting work for extending VAE towards Bayesian non-parametric! 

I wonder what is the motivation of constraining \pi to be positive and summed up to one (via taking the stick-breaking process)? 

I would suggest to have a comparison with our work "Variational Auto-encoded Deep Gaussian Processes". It is another Bayesian non-parametric VAE via using Gaussian process as the decoder,

[Official Review · AnonReviewer3 · rating 8 · confidence 5 · 16 Dec 2016]
**Good paper**

Summary: This is the first work to investigate stick-breaking priors, and corresponding inference methods, for use in VAEs. The background material is explained clearly, as well as the explanation of the priors and posteriors and their DNCP forms. The paper is really well written.

In experiments, they find that stick-breaking priors does not generally improve upon spherically Gaussian priors in the completely unsupervised setting, when measured w.r.t. log-likelihood. The fact that they do report this 'negative' result suggests good scientific taste. In a semi-supervised setting, the results are better.

Comments:
- sec 2.1: There is plenty of previous work with non-Gaussian p(z): DRAW, the generative ResNet paper in the IAF paper, Ladder VAEs, etc.
- sec 2.2: two comma's
- text flow eq 6: please refer to appendix with the closed-form KL divergence
- "The v's are sampled via" => "In the posterior, the v's are sampled via". It's not clear you're talking about the posterior here, instead of the prior.
- The last paragraph of section 4 is great.
- Sec 7.1: "Density estimation" => Technically you're also doing mass estimation.
- Sec 7.1: 100 IS samples is a bit on the low side. 
- Figure 3(f). Interesting that k-NN works so well on raw pixels.

[Official Review · AnonReviewer1 · rating 8 · confidence 4 · 17 Dec 2016]
**Motivation is not very clear but good paper overall**

This paper presents an approach which modifies the variational auto-encoder (VAE) framework so as to use stochastic latent dimensionality. This is achieved by using an inherently infinite prior, the stick-breaking process. This is coupled with inference tailored to this model, specifically the Kumaraswamy distribution as an approximate variational posterior. The resulting model is named the SB-VAE which also has a semi-supervised extension, in similar vein to the original VAE paper.

There's a lot of interest in VAEs these days; many lines of work seek to achieve automatic "black-box" inference in these models. For example, the authors themselves mention parallel work by Blei's lab (also others) towards this direction. However, there's a lot of merit in investigating more bespoke solutions to new models, which is what the authors are doing in this paper. Indeed, a (useful) side-effect of providing efficient inference for the SB-VAE is drawing attention to the use of the Kumaraswamy distribution which hasn't been popular in ML.

Although the paper is in general well structured, I found it confusing at parts. I think the major source of confusion comes from the fact that the model specification and model inference are discussed in a somehow mixed manner. The pre-review questions clarified most parts.

I have two main concerns regarding the methodology and motivation of this paper. Firstly, conditioning the model directly on the stick-breaking weights seems a little odd. I initially thought that there was some mixture probabilistic model involved, but this is not the case. To be fair, the authors discuss about this issue (which became clearer to me after the pre-review questions), and explain that they're investigating the apparently challenging problem of using a base distribution G_0. The question is whether their relaxation is still useful. From the experiments it seems that the method is at least competitive, so the answer is yes. Hopefully an extension will come in the future, as the authors mention.

The second concern is about the motivation of this method. It seems that the paper fails to clearly explain in a convincing way why it is beneficial to reformulate the VAE as a SB-VAE. I understand that the non-parametric property induced by the prior might result in better capacity control, however I feel that this advantage (and potentially others which are still unclear to me) is not sufficiently explained and demonstrated. Perhaps some comparison with a dropout approach or a more thorough discussion related to dropout would make this clearer.

Overall, I found this to be an interesting paper, it would be a good fit for ICLR.

[Final Decision · Program Chairs · 06 Feb 2017]
**ICLR committee final decision**

This paper will make a positive contribution to the conference, especially since it is one of the first to look at stick-breaking as it applies to deep generative models. The paper will make a positive contribution to the conference.